# Barriers to healthcare access and healthcare seeking for childhood illnesses among childbearing women in sub-Saharan Africa: A multilevel modelling of Demographic and Health Surveys

Bright Opoku Ahinkorah[1], Eugene Budu[2]*, Abdul-Aziz Seidu[2,3], Ebenezer Agbaglo[4], Collins Adu[5], Edward Kwabena Ameyaw[1], Irene Gyamfuah Ampomah[3], Anita Gracious Archer[6], Kwaku Kissah-Korsah[2], Sanni Yaya[7,8]

1 School of Public Health, Faculty of Health, University of Technology Sydney, Sydney, Australia, 2 Department of Population and Health, University of Cape Coast, Cape Coast, Ghana, 3 College of Public Health, Medical and Veterinary Services, James Cook University, Townsville, Australia, 4 Department of English, University of Cape Coast, Cape Coast, Ghana, 5 Department of Health Promotion, and Disability Studies, Kwame Nkrumah University of Science and Technology, Kumasi, Ghana, 6 School of Nursing and Midwifery, University of Health and Allied Sciences, Ho, Ghana, 7 School of International Development and Global Studies, University of Ottawa, Ottawa, Canada, 8 The George Institute for Global Health, The University of Oxford, Oxford, United Kingdom

* budueugene@gmail.com

**Data Availability Statement:** All relevant data are publicly accessible on the DHS Program website via the following URLs: Angola (2015-16): https://

## Abstract

### Introduction

The success of current policies and interventions on providing effective access to treatment for childhood illnesses hinges on families' decisions relating to healthcare access. In sub-Saharan Africa (SSA), there is an uneven distribution of child healthcare services. We investigated the role played by barriers to healthcare accessibility in healthcare seeking for childhood illnesses among childbearing women in SSA.

### Materials and methods

Data on 223,184 children under five were extracted from Demographic and Health Surveys of 29 sub-Saharan African countries, conducted between 2010 and 2018. The outcome variable for the study was healthcare seeking for childhood illnesses. The data were analyzed using Stata version 14.2 for windows. Chi-square test of independence and a two-level multivariable multilevel modelling were carried out to generate the results. Statistical significance was pegged at p<0.05. We relied on 'Strengthening the Reporting of Observational Studies in Epidemiology' (STROBE) statement in writing the manuscript.

### Results

Eighty-five percent (85.5%) of women in SSA sought healthcare for childhood illnesses, with the highest and lowest prevalence in Gabon (75.0%) and Zambia (92.6%) respectively. In

dhsprogram.com/data/dataset/Angola_Standard-DHS_2015.cfm?flag=0 Burkina Faso (2010): https://dhsprogram.com/data/dataset/Burkina-Faso_Standard-DHS_2010.cfm?flag=0 Benin (2017-18): https://dhsprogram.com/data/dataset/Benin_Standard-DHS_2017.cfm?flag=0 Burundi (2016-17): https://dhsprogram.com/data/dataset/Burundi_Standard-DHS_2016.cfm?flag=0 Congo DR. (2013-14): https://dhsprogram.com/data/dataset/Congo-Democratic-Republic_Standard-DHS_2013.cfm?flag=0 Congo (2011-12): https://dhsprogram.com/data/dataset/Congo_Standard-DHS_2011.cfm?flag=0 Cote D'Ivoire (2011-12): https://dhsprogram.com/data/dataset/Cote-d-Ivoire_Standard-DHS_2012.cfm?flag=0 Cameroon (2018): https://dhsprogram.com/data/dataset/Cameroon_Standard-DHS_2018.cfm?flag=0 Ethiopia (2016): https://dhsprogram.com/data/dataset/Ethiopia_Standard-DHS_2016.cfm?flag=0 Gabon (2012): https://dhsprogram.com/data/dataset/Gabon_Standard-DHS_2012.cfm?flag=0 Ghana (2014): https://dhsprogram.com/data/dataset/Ghana_Standard-DHS_2014.cfm?flag=0 Gambia (2013): https://dhsprogram.com/data/dataset/Gambia_Standard-DHS_2013.cfm?flag=0 Guinea (2018): https://dhsprogram.com/data/dataset/Guinea_Standard-DHS_2018.cfm?flag=0 Kenya (2014): https://dhsprogram.com/data/dataset/Kenya_Standard-DHS_2014.cfm?flag=0 Comoros (2012): https://dhsprogram.com/data/dataset/Comoros_Standard-DHS_2012.cfm?flag=0 Liberia (2013): https://dhsprogram.com/data/dataset/Liberia_Standard-DHS_2013.cfm?flag=0 Lesotho (2014): https://dhsprogram.com/data/dataset/Lesotho_Standard-DHS_2014.cfm?flag=0 Mali (2018): https://dhsprogram.com/data/dataset/Mali_Standard-DHS_2018.cfm?flag=0 Malawi (2015-16): https://dhsprogram.com/data/dataset/Malawi_Standard-DHS_2015.cfm?flag=0 Nigeria (2018): https://dhsprogram.com/data/dataset/Nigeria_Standard-DHS_2018.cfm?flag=0 Namibia (2013): https://dhsprogram.com/data/dataset/Namibia_Standard-DHS_2013.cfm?flag=0 Rwanda (2014-15): https://dhsprogram.com/data/dataset/Rwanda_Standard-DHS_2015.cfm?flag=0 Sierra Leone (2013): https://dhsprogram.com/data/dataset/Sierra-Leone_Standard-DHS_2013.cfm?flag=0 Senegal (2010-11): https://dhsprogram.com/data/dataset/Senegal_Standard-DHS_2010.cfm?flag=0 Chad (2014-15): https://dhsprogram.com/data/dataset/Chad_Standard-DHS_2014.cfm?flag=0 Togo (2013-14): https://dhsprogram.com/data/dataset/Togo_Standard-DHS_2013.cfm?flag=0 Uganda (2016): https://dhsprogram.com/data/dataset/Uganda_Standard-DHS_2016.cfm?flag=0 Zambia (2018-19): https://dhsprogram.com/data/dataset/Zambia_Standard-DHS_2018.cfm?flag=0

terms of the barriers to healthcare access, we found that women who perceived getting money for medical care for self as a big problem [AOR = 0.81 CI = 0.78–0.83] and considered going for medical care alone as a big problem [AOR = 0.94, CI = 0.91–0.97] had lower odds of seeking healthcare for their children, compared to those who considered these as not a big problem. Other factors that predicted healthcare seeking for childhood illnesses were size of the child at birth, birth order, age, level of community literacy, community socio-economic status, place of residence, household head, and decision-maker for healthcare.

## Conclusion

The study revealed a relationship between barriers to healthcare access and healthcare seeking for childhood illnesses in sub-Saharan Africa. Other individual and community level factors also predicted healthcare seeking for childhood illnesses in sub-Saharan Africa. This suggests that interventions aimed at improving child healthcare in sub-Saharan Africa need to focus on these factors.

## Introduction

Although the global prevalence of child deaths remains high, the world has made enormous progress in child survival in the past few decades. In 2018, an estimated 6.2 million children died under the age of 15 years across the globe [1]. Globally, 85% of deaths among children in 2018 occurred in the first five years of life, accounting for 5.3 million deaths [2]. Children continue to face widespread regional disparities in their chances of survival. Sub-Saharan Africa (SSA) remains the region with the highest child mortality in the world [1, 2]. In 2018, SSA recorded an average under-five mortality rate of 78 deaths per 1,000 live births, which translates to 1 in 13 children dying before their fifth birthday [2, 3]. Children in SSA are more than 16 times more likely to die before the age of 5, as compared to children in high-income countries [2]. In 2018, more than half of children under the age of 15 died in SSA [2].

The international community has made efforts to recognize the urgent need to end preventable child deaths, making it an important part of global child survival goals, the Sustainable Development Goals (SDGs), and the United Nations Global Strategy for Women's, Children's and Adolescents' Health (2016–2030). For instance, the Sustainable Development Goal (SDG) 3 aims at ensuring a reduction in deaths of children under-five to 25 deaths per 1,000 live births and deaths of newborns to 12 deaths per 1,000 live births throughout the world by 2030 [3]. This attention given to child health by the international community is important, given that children constitute a vulnerable population who deserve public health attention such as enhancing access to healthcare service [4]. Promoting access to cost-effective child health intervention is central to child mortality reduction [5].

With current policies and interventions on providing effective access to treatment of childhood illness, the success of the policies and interventions hinges on families' decision relating to whether to access healthcare service [6] and where to seek healthcare [7]. These decisions are extremely determined by availability of healthcare services. Therefore, increased access to essential child health services remains an integral effort to achieve SDG 3. The introduction of the Integrated Management of Childhood Illness (IMCI) has helped to reduce child mortality. It represents the main approach to promoting access to healthcare across a continuum of care,

Zimbabwe (2015): https://dhsprogram.com/data/dataset/Zimbabwe_Standard-DHS_2015.cfm?flag=0.

**Funding:** The author(s) received no specific funding for this work.

**Competing interests:** The authors have declared that no competing interests exist.

including a community and health system focus through its interrelated components, namely, clinical/facility IMCI, community IMCI, and health systems strengthening [8].

In SSA, there is an uneven distribution of child healthcare services [8]. Previous studies show that mothers in SSA usually do not have enough knowledge to identify danger signs or appropriate treatment to be given to their children to promote or maintain their health [4, 9, 10]. Besides, some mothers use traditional treatment ahead of modern healthcare services [9, 11]. Previous country-specific studies in Malawi [8], Ethiopia [4], and Nigeria [12] have revealed some barriers to health-seeking behaviour for childhood illnesses. A study by Bennett et al. [13], for example, identified child age, child sex, maternal education, place of residence, socio-economic status, and distance to the nearest healthcare facility as socio-demographic factors related to healthcare seeking for childhood illness. These factors could also serve as barriers to healthcare access and healthcare seeking for childhood illnesses. However, since those studies each focused on single countries, they failed to provide a panoramic view of the phenomenon. In view of this, this current study investigates the barriers to healthcare access and healthcare seeking for childhood illnesses among childbearing women in the entire SSA. Findings from this study will be potentially useful in contributing to efforts to promote child health, survival, and development in the sub-region.

## Materials and methods

### Data source

We pooled data from the Demographic and Health Surveys (DHSs) of 29 SSA countries, conducted between 2010 and 2018. Specifically, we used data from the children's files from the various countries. All women whose data are captured in this file are either caregivers of children under five or gave birth within the five years preceding the surveys. The DHS is a nationally representative survey that is conducted in over 85 low- and middle-income countries globally. The survey focuses on essential maternal and child health markers, including health seeking behaviour, contraceptive use, skilled birth attendance, immunization among under-fives, and intimate partner violence [14]. The survey employs a two-stage stratified sampling technique, which makes the data nationally representative. The study by Aliaga and Ruilin [15] provides details of the sampling process. Sample sizes are determined by the number of women in the selected households who fall within the ages 15–49 years for women and 15–64 years for men. Various quality control measures are employed to collect quality data. For example, consistency across the various countries is maintained by employing the same variables and measures (instruments). Nonetheless, countries are allowed to add specific variables of interest to suit their context. The survey staff are trainees who are instructed in standard DHS procedures, including general interviewing techniques, conducting interviews at the household level, and review of each question and mock interviews between participants. The DHSs in sub-Saharan Africa are usually conducted in English, French, and Portuguese depending on the official language of the country. To ensure participants comprehended/understood the questions being asked, the definitive questionnaires are first prepared in the official language in the specific country and subsequently translated into the major local languages at the various data collection points [14, 15]. In this study, we analysed data for a weighted sample of 223,184 children under five years who were alive during the surveys. Table 1 provides details of the countries, survey years, and samples used for the study. In this study, we relied on the 'Strengthening the Reporting of Observational Studies in Epidemiology' (STROBE) statement in writing the manuscript [16].

**Table 1. Distribution of the study sample by country.**

| Country | Survey year | Sample used | % of sample used |
|---------|-------------|-------------|------------------|
| Angola | 2015–16 | 5472 | 2.5 |
| Burkina Faso | 2010 | 14869 | 6.7 |
| Benin | 2017–18 | 12479 | 5.6 |
| Burundi | 2016–17 | 12448 | 5.6 |
| Congo DR. | 2013–14 | 15722 | 7.0 |
| Congo | 2011–12 | 3716 | 1.7 |
| Cote D'Ivoire | 2011–12 | 6113 | 2.7 |
| Cameroon | 2011 | 3025 | 1.4 |
| Ethiopia | 2016 | 10371 | 4.7 |
| Gabon | 2012 | 611 | 0.3 |
| Ghana | 2014 | 4951 | 2.2 |
| Gambia | 2013 | 7336 | 3.3 |
| Guinea | 2018 | 7462 | 3.3 |
| Kenya | 2014 | 7769 | 3.5 |
| Comoros | 2012 | 2700 | 1.2 |
| Liberia | 2013 | 4911 | 2.2 |
| Lesotho | 2014 | 964 | 0.4 |
| Mali | 2018 | 9671 | 4.3 |
| Malawi | 2015–16 | 14757 | 6.6 |
| Nigeria | 2018 | 23285 | 10.4 |
| Namibia | 2013 | 2034 | 0.9 |
| Rwanda | 2014–15 | 6624 | 3.0 |
| Sierra Leone | 2013 | 2589 | 1.2 |
| Senegal | 2010–11 | 10960 | 4.9 |
| Chad | 2014–15 | 5790 | 2.6 |
| Togo | 2013–14 | 3277 | 1.5 |
| Uganda | 2016 | 12783 | 5.7 |
| Zambia | 2018–19 | 4931 | 2.2 |
| Zimbabwe | 2015 | 5564 | 2.5 |
| Total | | 223,184 | 100.0 |

## Definition of variables

**Outcome variable.** The outcome variable for the study was healthcare seeking for childhood illnesses. It was derived as a composite variable from two questions, "Did [NAME] receive treatment for diarrhea?" and "Did [NAME] receive treatment from fever/cough?" The responses were "Yes" and "No". For the purpose of this study, respondents who answered "Yes" to any of the two questions were considered as seeking healthcare for childhood illnesses and were put in the category "Yes" and coded 1. On the other hand, those who answered "No" to the two questions were considered as those who did not seek healthcare for childhood illnesses and were put in the category "No" and coded 0.

**Independent variables.** The study considered barriers to accessing healthcare as the independent variables. These variables were generated by asking women whether they had serious problems in accessing healthcare for themselves when they are sick, by type of problem. The problems were difficulty with distance to health facility, difficulty in getting money for treatment, difficulty with getting permission to visit health facility, and difficulty in not wanting to go for medical help alone. In each of these instances, these variables were recoded as "Big problem" and "Not a big problem."

**Control variables.** Sixteen control variables consisting of four child factors (size of child at birth, birth order, twin status, and sex of child), eight maternal factors (age, marital status, employment, religion, parity, frequency of reading newspaper/magazine, frequency of listening to radio, and frequency of watching television), and five community factors (healthcare decision-making capacity, place of residence, community literacy level, community socio-economic status, and sex of household head) were considered in our study. Child and maternal factors were combined as individual factors. The selection of these variables was influenced by their relevance in previous studies on health-seeking for childhood illnesses [8, 17–19]. The categories generated for each of these variables can be found in Table 2.

**Table 2. Distribution of barriers to healthcare, individual and community factors and healthcare seeking for childhood illnesses among childbearing women in sub-Saharan Africa.**

| Variable | Weighted N | Weighted % | Healthcare seeking | χ2 (p-value) |
|---|---|---|---|---|
| **Getting permission for medical care for self** | | | | 127.0 (<0.001) |
| Big problem | 43762 | 19.6 | 83.8 | |
| Not a big problem | 179422 | 80.4 | 86.9 | |
| **Getting money for medical care for self** | | | | 558.5 (<0.001) |
| Big problem | 125809 | 56.4 | 83.9 | |
| Not a big problem | 97375 | 43.6 | 87.5 | |
| **Distance to facility for medical care for self** | | | | 265.3 (<0.001) |
| Big problem | 92234 | 41.3 | 84.1 | |
| Not a big problem | 130950 | 58.7 | 86.5 | |
| **Wanting to go for medical care alone** | | | | 185.2 (<0.001) |
| Big problem | 50274 | 22.5 | 83.6 | |
| Not a big problem | 172909 | 77.5 | 86.0 | |
| **Community literacy level** | | | | 154.7 (<0.001) |
| Low | 77145 | 34.6 | 84.9 | |
| Medium | 69376 | 31.1 | 84.7 | |
| High | 76661 | 34.3 | 86.8 | |
| **Community socio-economic status** | | | | 243.9 (<0.001) |
| Low | 122650 | 54.9 | 84.6 | |
| Medium | 25337 | 11.4 | 85.0 | |
| High | 75196 | 33.7 | 87.2 | |
| **Residence** | | | | 202.3 (<0.001) |
| Urban | 61987 | 27.8 | 87.2 | |
| Rural | 161197 | 72.2 | 84.8 | |
| **Age** | | | | 118.1 (<0.001) |
| 15–19 | 9865 | 4.4 | 82.2 | |
| 20–24 | 46744 | 20.9 | 84.9 | |
| 25–29 | 62222 | 27.9 | 85.6 | |
| 30–34 | 49889 | 22.4 | 85.8 | |
| 35–39 | 34051 | 15.3 | 86.2 | |
| 40–44 | 15676 | 7.0 | 85.6 | |
| 45–49 | 4737 | 2.1 | 85.6 | |
| **Marital status** | | | | 296.4 (<0.001) |
| Married | 183738 | 82.3 | 86.1 | |
| Cohabiting | 39446 | 17.7 | 82.7 | |
| **Healthcare decision-making capacity** | | | | 9.6 (<0.05) |
| Alone | 34435 | 15.4 | 84.9 | |

*(Continued)*

**Table 2.** (Continued)

| Variable | Weighted N | Weighted % | Healthcare seeking | χ2 (p-value) |
|---|---|---|---|---|
| Not alone | 188749 | 84.6 | 85.5 | |
| **Parity** | | | | 130.1 (<0.001) |
| One birth | 24725 | 11.1 | 83.4 | |
| Two births | 42493 | 19.0 | 86.4 | |
| Three births | 39052 | 17.5 | 86.3 | |
| Four or more births | 116914 | 52.4 | 85.5 | |
| **Employment status** | | | | 1.2 (0.269) |
| Not working | 56674 | 25.4 | 85.3 | |
| Working | 166510 | 74.6 | 85.5 | |
| **Religion** | | | | 161.7 (<0.001) |
| Christianity | 136422 | 61.1 | 85.0 | |
| Islam | 76937 | 34.5 | 86.6 | |
| Traditionalist | 4712 | 2.1 | 84.0 | |
| No religion | 5113 | 2.3 | 82.0 | |
| **Frequency of reading newspaper** | | | | 65.7 (<0.001) |
| Not at all | 197283 | 88.4 | 85.3 | |
| Less than once a week | 16086 | 7.2 | 86.6 | |
| At least once a week | 9815 | 4.4 | 87.9 | |
| **Frequency of listening to radio** | | | | 180.2 (<0.001) |
| Not at all | 98150 | 44.0 | 84.4 | |
| Less than once a week | 44712 | 20.0 | 86.3 | |
| At least once a week | 80322 | 36.0 | 86.4 | |
| **Frequency of watching television** | | | | 306.2 (<0.001) |
| Not at all | 148268 | 66.4 | 84.6 | |
| Less than once a week | 28389 | 12.7 | 86.9 | |
| At least once a week | 46526 | 20.9 | 87.6 | |
| **Sex of household head** | | | | 10.1 (<0.05) |
| Male | 191890 | 86.0 | 85.5 | |
| Female | 31294 | 14.0 | 84.9 | |
| **Size of child at birth** | | | | 242.4 (<0.001) |
| Larger than average | 81851 | 36.7 | 85.1 | |
| Average | 104566 | 46.8 | 86.5 | |
| Smaller than average | 36767 | 16.5 | 83.3 | |
| **Birth order** | | | | 80.6 (<0.001) |
| First | 41576 | 18.6 | 86.3 | |
| 2–4 | 108425 | 48.6 | 85.7 | |
| 5+ | 73182 | 32.8 | 84.5 | |
| **Twin status** | | | | 49.8 (<0.001) |
| Single birth | 214929 | 96.3 | 85.3 | |
| Multiple birth | 8255 | 3.7 | 881 | |
| **Sex of child** | | | | 0.9 (0.337) |
| Male | 112929 | 50.6 | 85.4 | |
| Female | 110255 | 49.4 | 85.5 | |
| N | 223184 | | | |

## Statistical analyses

The data were analyzed using Stata version 14.2 for windows. The datasets were extracted from each country's datafiles, cleaned, and recoded. The recoding was done to ensure consistency in the variables across the countries. After that, the dataset was appended to generate pooled data [14]. The analyses began with the computation of the prevalence of healthcare seeking for childhood illnesses using bar chart. This was followed by the distribution of healthcare seeking for childhood illnesses across the barriers to healthcare, child, maternal, and community level factors. Chi-square test of independence was used to assess the statistical significance of the association between each of the factors and healthcare seeking for childhood illnesses at a p-value of 0.05 (see Table 1). Next, a two-level multivariable logistic regression analysis was carried out to examine the influence of barriers to healthcare access and healthcare seeking for childhood illnesses while controlling for the effect of individual and community factors. The two-level modelling in this study implies that women were nested within clusters (primary sampling units). Clusters were considered as random effects to cater for the unexplained variability at the community level [20].

In terms of the modelling, four models were fitted and they comprised the empty model (model 0), Model I (individual factors and barriers to healthcare access), Model II (community level factors only), and Model III (all factors). Model 0 showed the variance in the outcome variable that is attributed to the clustering of the primary sampling units (PSUs) without the explanatory variables. The Stata command "melogit" was used in fitting these models. Model comparison was done using the log-likelihood ratio (LLR) and Akaike's information criterion (AIC) tests. The highest log-likelihood and the lowest AIC were used to determine the best fit model (see Table 3). Odds ratio and associated 95% confidence intervals (CIs) were presented for all the models apart from Model 0 (see Table 2). To check for high correlation among the explanatory variables, a test for multicollinearity was carried out using the variance inflation factor (VIF), and the results showed no evidence of high collinearity (Mean VIF = 1.51, Maximum VIF = 3.18, and Minimum VIF = 1.02). Sample weight (v005/1,000,000) and SVY command were used to correct for over- and under-sampling, and the complex survey design and generalizability of the findings respectively.

## Ethical approval

Ethical clearance was obtained from the Ethics Committee of ORC Macro Inc. as well as Ethics Boards of partner organizations of the various countries such as the Ministries of Health. The DHS follows the standards for ensuring the protection of respondents' privacy. Inner City Fund (ICF) International ensured that the survey complies with the U.S. Department of Health and Human Services regulations for the respect of human subjects. The survey indicates that the respondents provided both written and oral consent prior to the data collection. However, this was a secondary analysis of data and, therefore, no further approval was required since the data is available in the public domain. Further information about the DHS data usage and ethical standards are available at http://goo.gl/ny8T6X.

## Results

### Prevalence of health seeking for childhood illness in sub-Saharan Africa

As shown in Fig 1, approximately eighty-six percent (85.5%) of women in SSA sought healthcare for childhood illnesses, with the highest and lowest prevalence in Gabon (75.0%) and Zambia (92.6%) respectively.

**Table 3. Mixed effects results on barriers to healthcare, individual and community factors associated with healthcare seeking for childhood illnesses among child-bearing women in sub-Saharan Africa.**

| Variable | Model 0 | Model I AOR[95%CI] | Model II AOR[95%CI] | Model III AOR[95%CI] |
|---|---|---|---|---|
| **Size of child at birth** | | | | |
| Larger than average | | 0.91*** (0.89–0.93) | | 0.91*** (0.88–0.93) |
| Average | | 1 | | 1 |
| Smaller than average | | 0.78***(0.75–0.80) | | 0.78***(0.75–0.80) |
| **Birth order** | | | | |
| First | | 1 | | 1 |
| 2–4 | | 0.54*** (0.51–0.58) | | 0.54***(0.51–0.58) |
| 5+ | | 0.43*** (0.40–0.46) | | 0.43***(0.40–0.46) |
| **Twin status** | | | | |
| Single birth | | 0.76*** (0.71–0.81) | | 0.76***(0.71–0.81) |
| Multiple birth | | 1 | | 1 |
| **Age** | | | | |
| 15–19 | | 0.66***(0.61–0.71) | | 0.67***(0.62–0.72) |
| 20–24 | | 0.76***(0.72–0.80) | | 0.76***(0.72–0.80) |
| 25–29 | | 0.83***(0.79–0.86) | | 0.83***(0.79–0.87) |
| 30–34 | | 0.90***(0.87–0.94) | | 0.90*** (0.87–0.94) |
| 35–39 | | 1 | | 1 |
| 40–44 | | 1.00(0.95–1.06) | | 1.01(0.85–1.06) |
| 45–49 | | 1.02 (0.94–1.11) | | 1.02 (0.94–1.12) |
| **Marital status** | | | | |
| Married | | 1 | | 1 |
| Cohabiting | | 0.84*** (0.81–0.86) | | 0.84*** (0.81–0.86) |
| **Parity** | | | | |
| One birth | | 0.55***(0.52–0.59) | | 0.55***(0.52–0.59) |
| Two births | | 1 | | 1 |
| Three births | | 1.12***(1.07–1.17) | | 1.12***(1.07–1.17) |
| Four or more births | | 1.13***(1.08–1.18) | | 1.14***(1.09–1.19) |
| **Religion** | | | | |
| Christianity | | 0.86***(0.86–0.91) | | 0.89***(0.87–0.92) |
| Islam | | 1 | | 1 |
| Traditionalist | | 0.87***(0.80–0.94) | | 0.87***(0.81–0.95) |
| No religion | | 0.78***(0.73–0.85) | | 0.79***(0.74–0.86) |
| **Frequency of reading newspaper** | | | | |
| Not at all | | 0.91**(0.85–0.98) | | 0.92*(0.86–0.99) |
| Less than once a week | | 0.90* (0.83–0.98) | | 0.91*(0.84–0.98) |
| At least once a week | | 1 | | 1 |
| **Frequency of listening to radio** | | | | |
| Not at all | | 0.92***(0.89–0.96) | | 0.93***(0.89–0.96) |
| Less than once a week | | 1 | | 1 |
| At least once a week | | 0.97(0.94–1.01) | | 0.97(0.94–1.01) |
| **Frequency of watching television** | | | | |
| Not at all | | 0.89***(0.86–0.92) | | 0.92***(0.89–0.96) |
| Less than once a week | | 0.98(0.93–1.03) | | 1.00(0.96–1.05) |
| At least once a week | | 1 | | 1 |
| **Getting permission for medical care for self** | | | | |
| Big problem | | 0.99 (0.96–1.03) | | 0.99 (0.96–1.02) |

*(Continued)*

**Table 3.** (Continued)

| Variable | Model 0 | Model I AOR[95%CI] | Model II AOR[95%CI] | Model III AOR[95%CI] |
|---|---|---|---|---|
| Not a big problem | | 1 | | 1 |
| **Getting money for medical care for self** | | | | |
| Big problem | | 0.81***(0.78–0.83) | | 0.81***(0.78–0.83) |
| Not a big problem | | 1 | | 1 |
| **Distance to facility for medical care for self** | | | | |
| Big problem | | 0.97* (0.91–0.97) | | 0.98 (0.95–1.01) |
| Not a big problem | | 1 | | 1 |
| **Wanting to go for medical care alone** | | | | |
| Big problem | | 0.94***(0.91–0.97) | | 0.94***(0.91–0.97) |
| Not a big problem | | 1 | | 1 |
| **Community literacy level** | | | | |
| Low | | | 0.97(0.93–1.00) | 1.01(0.98–1.05) |
| Medium | | | 0.93***(0.90–0.96) | 0.95**(0.92–0.99) |
| High | | | 1 | 1 |
| **Community socio-economic status** | | | | |
| Low | | | 0.87***(0.84–0.91) | 0.95**(0.92–0.99) |
| Medium | | | 0.89***(0.84–0.93) | 0.94**(0.89–0.98) |
| High | | | 1 | 1 |
| **Residence** | | | | |
| Urban | | | 1 | 1 |
| Rural | | | 0.90***(0.87–0.94) | 0.95**(0.91–0.98) |
| **Healthcare decision-making** | | | | |
| Alone | | | 0.95***(0.92–0.98) | 0.96**(0.92–0.99) |
| Not alone | | | 1 | 1 |
| **Sex of household head** | | | | |
| Male | | | 1 | 1 |
| Female | | | 0.94***(0.91–0.98) | 0.96**(0.92–0.99) |
| **Random effect result** | | | | |
| PSU variance (95% CI) | 0.05 (0.04–0.06) | 0.04 (0.04–0.05) | 0.05(0.04–0.06) | 0.04(0.03–0.05) |
| ICC | 0.015 | 0.013 | 0.014 | 0.013 |
| LR Test | χ2 = 406.68, p = 0.000 | χ2 = 360.24, p = 0.000 | χ2 = 387.93, p = 0.0000 | χ2 = 356.02, p = 0.000 |
| Wald chi-square | Reference | 2077.59 | 312.75 | 2144.42 |
| Model fitness | | | | |
| Log-likelihood | -92397.577 | -91320.966 | -92237.854 | -91284.861 |
| AIC | 184799.2 | 182701.9 | 184493.7 | 182643.7 |
| N | 223184 | 223184 | 223184 | 223184 |

Exponentiated coefficients; 95% confidence intervals in brackets

* $p < 0.05$

** $p < 0.01$

*** $p < 0.001$

N = Sample size; 1 = Reference category; PSU = Primary Sampling Unit; ICC = Intra-Class Correlation; LR Test = Likelihood ratio Test; AIC = Akaike's Information Criterion

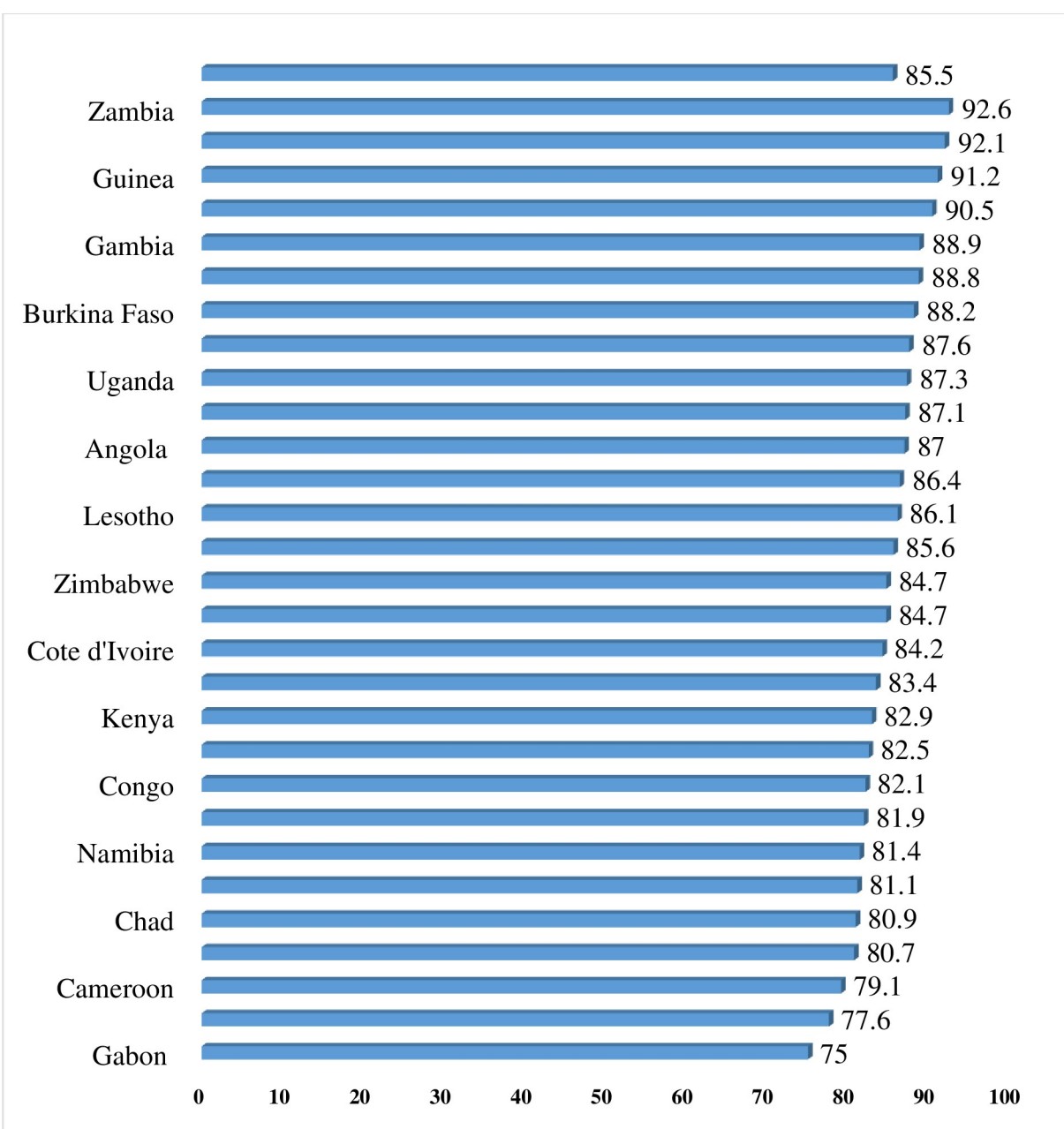

**Fig 1. Prevalence of healthcare seeking for childhood illness in sub-Saharan Africa.**

### Distribution of barriers to healthcare, individual, and community factors and healthcare seeking for childhood illnesses among childbearing women in sub-Saharan Africa

Table 2 shows the distribution of barriers to healthcare, individual, and community factors, and healthcare seeking for childhood illnesses among childbearing women in SSA. It was found that 19.6% of the childbearing women indicated that they have a big problem getting permission to seek medical care, 56.4% indicated they face a big problem getting money to seek care, while 41.3% and 22.5% also said it is a big problem for them when it comes to

distance and not wanting to go to a health facility alone respectively. About 46.8% of the children were of average size at birth, 48.6% were parity 2–4, slightly above ninety-six percent (96.3%) of them were single births, and 50.6% were males. The study also showed that 27.9% of the mothers were aged 25–29, fifty-two percent (52%) had four or more births, 74.6% were working, and 61.1% were Christians. In terms of exposure to mass media, the study revealed that 44%, 66.4%, and 34.6% did not read newspaper, listen to radio, or watch television at all. The study also showed that 54.9% were in low category of community socio-economic status, 72.2% resided in rural areas, 86.0% were in male-headed households, and 84.6% did not take healthcare decisions alone. The chi-square analysis showed that, except employment status and sex of child, all the independent variables had statistically significant association with healthcare seeking for childhood illnesses at $p < 0.05$.

## Barriers to healthcare, individual, and community factors associated with healthcare seeking for childhood illnesses among childbearing women in sub-Saharan Africa

Table 3 shows the results on barriers to healthcare, individual, and community factors associated with healthcare seeking for childhood illnesses among childbearing women in SSA. In the fully adjusted model, controlling for the effects of individual and community factors, it was found that women who perceived getting money for medical care for self as a big problem [AOR = 0.81 CI = 0.78–0.83] and considered going for medical care alone as a big problem [AOR = 0.94, CI = 0.91–0.97] had lower odds of seeking healthcare for their children, compared to those who considered these as not a big problem.

The results further showed that mothers with children larger [AOR = 0.91 CI = 0.88–0.93] and smaller than average [AOR = 0.78, CI = 0.75–0.80] had lower odds of seeking care for their children's illnesses, compared to those who had children with average size at birth. The odds of healthcare seeking for childhood illnesses reduced with high birth order, as mothers with children who were at birth order 5 or more were less likely to seek healthcare for childhood illnesses, compared with children of first birth order [AOR = 0.43, CI = 0.40–0.46]. Mothers of single birth children were less likely to seek healthcare for childhood illnesses, compared to those with multiple births [AOR = 0.76, CI = 0.71–0.81].

The likelihood of healthcare seeking for childhood illnesses reduced with age, as mothers aged 15–19 had the lowest odds of seeking healthcare for their children's illnesses, compared with those aged 45–49 [AOR = 0.67, CI = 0.62–0.72]. The results also showed that mothers who were cohabiting [AOR = 0.84, CI = 0.81–0.86], those with parity one [AOR = 0.55, CI = 0.52–0.59], and those with no religion [AOR = 0.79, CI = 0.74–0.86] had lower odds of seeking healthcare for their children's ailments, compared with married women, women with parity three, and Muslim women respectively. With mass media exposure, it was revealed that women who read newspaper/magazine less than once a week [AOR = 0.91, CI = 0.84–0.98), those who did not listen to radio at all [AOR = 0.93, CI = 0.89–0.96], and those who did not watch television at all [AOR = 0.92, CI = 0.89–0.96] had lower odds of seeking healthcare for their children's ailments.

The study further showed that those with medium level of community literacy [AOR = 0.95 CI = 0.92–0.99], mothers with medium socio-economic status [AOR = 0.94, CI = 0.89–0.98], those in rural areas [AOR = 0.95, CI = 0.91–0.98], those in female-headed households [AOR = 0.96, CI = 0.92–0.99], and those who take healthcare decisions alone [AOR = 0.96, CI = 0.92–0.99] had lower odds of seeking healthcare for childhood illnesses.

## Discussion

Though there have been some improvements in child survival globally, a considerable number of children still die in poor-income countries, especially in SSA [1]. This suggests the need to pay more attention to child healthcare services in SSA. In the present study, we investigated the barriers and other factors associated with health-seeking behavior for childhood illnesses in sub-Saharan Africa. This was done with the aim of providing target areas for interventions intended to improve child healthcare in the sub-region. The study revealed an overall health-seeking prevalence of 85%, with inter-country variations. In terms of specific countries, the prevalence ranged from 75.0% in Gabon to 92.6% in Zambia. This difference in prevalence of health-seeking for childhood diseases across countries could be a reflection of differences in socio-cultural factors across the countries. It may also suggest an uneven distribution of child healthcare services across the countries in the sub-region [8].

The study found a significant association between access to money and mothers' healthcare seeking behavior for their children's ailments. Specifically, women who had a big problem getting money had a lower likelihood to seek healthcare for their children's diseases, compared to their counterparts who had no problem getting money for healthcare. This finding confirms the results of Assefa et al. [21], who reported financial constraints as a barrier to health-seeking behaviour for childhood diseases in Northzoa Zone of Oromia Regional State of Ethiopia. Bedford and Sharkey [22] similarly reported financial problems as a barrier to heath-seeking behaviour for childhood diseases in Kenya, Niger, and Nigeria. Concurrently, Noordam et al. [23] also reported higher likelihood of health-seeking for childhood diseases among mothers from rich households. In SSA countries that have health insurance schemes, such as Ghana, Kenya, Rwanda, and Ethiopia [24], the barrier may not have resulted from financial charges that come with healthcare, but other indirect costs involved, such as the cost of transportation to health centres [25].

In the present study, we observed an association between mothers' age and the likelihood of seeking healthcare for their children's diseases. Specifically, older mothers recorded higher likelihoods of seeking healthcare for their children, compared to younger mothers. Related to this finding, the study also reveals that mothers with single births were less likely to seek care for their children's illnesses, compared with mothers with multiple births. Since multiple births come with old age, these findings collectively seem to suggest that older mothers are more experienced in determining when their children need healthcare, and this may be a reason for their higher probability of seeking healthcare for their children's illnesses [9]. This finding is, however, inconsistent with findings of a study by Astale and Chenault [26], who found higher likelihood of health-seeking behaviour for childhood diseases among younger mothers in Ethiopia. Gelaw et al. [27] also reported that younger mothers, relative to older mothers, were more likely to seek medical care for their children's diseases. On the contrary, a study by Amin et al. [28] in rural Bangladesh reported no significance association between mothers' age and health-seeking behaviour for childhood diseases.

Women with no religion had lower odds of seeking healthcare for their children's illnesses, compared with Muslim women. This finding seems to suggest that religion encourages health-seeking behavior for childhood illnesses. A study by Abdulkadir and Abdulkadir [29] similarly reported that being a votary of Muslim faith is a strong determinant of maternal health-seeking for childhood diseases. Mebratie et al. [30] also reported a significant association between religion and health-seeking behaviour for childhood illnesses. Some studies, however, found no association between religion and healthcare-seeking behaviour of mothers for childhood diseases [27]. Bedford and Sharkey [22], on the other hand, reported how some religious beliefs rather prohibited health-seeking behaviour for childhood diseases in Kenya and Nigeria.

Women who were not exposed to mass media reported lower likelihood of seeking health-care for their children's diseases, compared to those who were more exposed to newspapers, radio, and television. This finding affirms the findings of Yaya and Bishwajit [31], and Gebret-sadik et al. [32] in Uganda and Ethiopia, respectively. Adinan et al. [33] similarly reported that parents with mass media exposure were about two times more likely to seek healthcare for their children, as compared to those without mass media exposure. In explaining this finding, Gebredsadik et al. [32] and Adinan et al. [33] noted that mass media are sometimes used to create awareness on health issues; therefore, mothers' access to mass media increases their awareness of the importance of child healthcare, and this may explain their higher likelihood of seeking healthcare for their sick children. However, studies by Amin et al. [28], and Chand-wani and Pandor [34] in Bangladesh and India, respectively, reported no association between mass media exposure and mothers' health-seeking behaviour for their children's illnesses.

Rural-urban differentials also showed a significant association with mothers' health-seeking for their children's illnesses. Specifically, mothers in the rural areas reported lower likelihood of seeking care for their children's diseases, compared to mothers who dwelled in urban areas. Assefa et al. [21] and Gelaw et al. [27] reported a similar finding in Ethiopia. Similarly, Ferdous et al. [35] reported low health-seeking behaviour for childhood diseases among mothers in rural Bangladesh. This low health-seeking behavior for childhood diseases among mothers in rural areas, compared to urban areas, could be attributed to the disparity in the distribution of infrastructure that could facilitate healthcare. In most African countries, relative to urban areas, rural areas suffer infrastructure deficit, in terms of health facilities, good roads, and transportation networks [36–38], and this is likely to negatively influence the health-seeking behavior of residents of rural areas. Noordam et al. [23], however, found no significant associa-tion between rural-urban residency and health-seeking behaviour for childhood ailments.

## Strengths and limitations

The findings of the study need to be interpreted in the light of some strengths and limitations. The main strength of this study lies in the use of nationally representative data. The sampling strategy–multi-stage—employed by the DHS usually helps to minimize selection bias [39]. Also, the DHS employs standard data collection instruments and procedures. The DHS con-duct extensive training of interviewers, which guarantees the collection of reliable information from survey participants. With the sampling strategy and procedures, the findings of the study can be generalized to all children in the studied countries. In terms of analysis, the clustered nature of the data was accounted for by employing multi-level modelling. This ensured vigor-ous analysis of the data. Aside these strengths, the study also suffers some limitations that need to be acknowledged. In the first place, since the study relied on secondary data, our analysis was limited to only the variables available in the data. With this, other variables that could be significantly associated with health-seeking behaviour for childhood diseases might have been excluded from the analysis. The study might also suffer social desirability bias since some of the variables were self-reported. The possibility of recall biases that often characterize DHS data, due to the retrospective nature of the report, cannot be overruled.

## Conclusion

The study revealed a relationship between barriers to healthcare access and healthcare seeking for childhood illnesses in SSA. Other individual and community level factors also predicted healthcare seeking for childhood illnesses in SSA. This suggests that interventions aimed at improving child healthcare in SSA need to focus on these factors.

## Acknowledgments

We acknowledge Measure DHS for providing us with the data upon which the findings of this study were based.

## Author Contributions

**Conceptualization:** Bright Opoku Ahinkorah, Eugene Budu, Abdul-Aziz Seidu.

**Data curation:** Bright Opoku Ahinkorah, Eugene Budu, Abdul-Aziz Seidu.

**Formal analysis:** Bright Opoku Ahinkorah, Eugene Budu, Abdul-Aziz Seidu.

**Funding acquisition:** Bright Opoku Ahinkorah, Kwaku Kissah-Korsah.

**Investigation:** Bright Opoku Ahinkorah, Anita Gracious Archer, Kwaku Kissah-Korsah.

**Methodology:** Bright Opoku Ahinkorah, Eugene Budu, Abdul-Aziz Seidu, Kwaku Kissah-Korsah.

**Project administration:** Bright Opoku Ahinkorah, Anita Gracious Archer, Kwaku Kissah-Korsah.

**Resources:** Bright Opoku Ahinkorah, Anita Gracious Archer, Kwaku Kissah-Korsah.

**Software:** Bright Opoku Ahinkorah, Kwaku Kissah-Korsah.

**Supervision:** Bright Opoku Ahinkorah, Kwaku Kissah-Korsah, Sanni Yaya.

**Validation:** Bright Opoku Ahinkorah, Abdul-Aziz Seidu, Anita Gracious Archer, Kwaku Kissah-Korsah, Sanni Yaya.

**Visualization:** Bright Opoku Ahinkorah, Anita Gracious Archer, Kwaku Kissah-Korsah.

**Writing – original draft:** Bright Opoku Ahinkorah, Eugene Budu, Abdul-Aziz Seidu, Ebenezer Agbaglo, Collins Adu, Edward Kwabena Ameyaw, Irene Gyamfuah Ampomah, Anita Gracious Archer, Kwaku Kissah-Korsah, Sanni Yaya.

**Writing – review & editing:** Bright Opoku Ahinkorah, Eugene Budu, Abdul-Aziz Seidu, Ebenezer Agbaglo, Collins Adu, Edward Kwabena Ameyaw, Irene Gyamfuah Ampomah, Anita Gracious Archer, Kwaku Kissah-Korsah, Sanni Yaya.

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
