## [Decision Letter · Decision Letter 0]

18 Nov 2020

PONE-D-20-18533

Barriers to healthcare access and healthcare seeking for childhood illnesses among childbearing women in sub-Saharan Africa: A multilevel modelling of Demographic and Health Surveys

PLOS ONE

Dear Dr. Budu,

Thank you for submitting your manuscript to PLOS ONE. After careful consideration, we feel that it has merit but does not fully meet PLOS ONE’s publication criteria as it currently stands. Therefore, we invite you to submit a revised version of the manuscript that addresses the points raised during the review process.

We look forward to receiving your revised manuscript.

Kind regards,

Claudia Marotta

Academic Editor

PLOS ONE

Journal Requirements:

Additional Editor Comments (if provided):

Dear Authors follow reviewer suggestion to improve your article

Reviewers' comments:

Reviewer's Responses to Questions

**Comments to the Author**

1. Is the manuscript technically sound, and do the data support the conclusions?

Reviewer #1: Yes

Reviewer #2: Yes

2. Has the statistical analysis been performed appropriately and rigorously? 

Reviewer #1: Yes

Reviewer #2: Yes

3. Have the authors made all data underlying the findings in their manuscript fully available?

Reviewer #1: Yes

Reviewer #2: Yes

4. Is the manuscript presented in an intelligible fashion and written in standard English?

Reviewer #1: Yes

Reviewer #2: Yes

5. Review Comments to the Author

Reviewer #1: This manuscript analyzed data from Demographic and Health Surveys of 29 sub-Saharan African countries. The results showed that mothers’ age, religion, mass media exposure, and rural-urban residency are significantly associated with healthcare seeking for childhood illnesses in sub-Saharan Africa. The topic of this manuscript falls within the scope of Plus one. However, there are several major issues with the manuscript need to be addressed, as follows:

1. The data used in the manuscript come from different countries. Please indicate how the accuracy and quality control of the data was carried out.

2. Some of the independent variables in this study were selected by asking women for relevant information. Please indicate how to ensure the validity and reliability of the data.

3. Using STATA for a multilevel modelling may not be appropriate. Please explain the applicability of using STATA in this study.

Reviewer #2: After reading, I've got some questions/suggestions to the authors:

- There are too many lines in the tables, especially vertical lines, which do not conform to the statistical table specification.

- The quality of DHS data should be mentioned in the paper where appropriate or explained by reference.

6. PLOS authors have the option to publish the peer review history of their article (what does this mean?). If published, this will include your full peer review and any attached files.

Reviewer #1: No

Reviewer #2: No

---

## [Author Response · Author response to Decision Letter 0]

20 Nov 2020

AUTHOR’S RESPONSE TO REVIEWS

Barriers to healthcare access and healthcare seeking for childhood illnesses among childbearing women in sub-Saharan Africa: A multilevel modelling of Demographic and Health Surveys

Date: 20/11/2020

Dear Editor and Reviewers,

We are pleased to resubmit for publication the revised version of “Barriers to healthcare access and healthcare seeking for childhood illnesses among childbearing women in sub-Saharan Africa: A multilevel modelling of Demographic and Health Surveys”. We are extremely grateful to you for giving us the opportunity to revise and resubmit this manuscript. We appreciate the time and constructive feedback provided by the reviewers and the Editor. The manuscript has certainly benefited from these insightful suggestions. Based on the comments, we have responded specifically to each suggestion below. 

RESPONSE TO REVIEWS 

Reviewer #1: 

This manuscript analyzed data from Demographic and Health Surveys of 29 sub-Saharan African countries. The results showed that mothers’ age, religion, mass media exposure, and rural-urban residency are significantly associated with healthcare seeking for childhood illnesses in sub-Saharan Africa. The topic of this manuscript falls within the scope of PlOS one. However, there are several major issues with the manuscript need to be addressed, as follows:

1. Comment: The data used in the manuscript come from different countries. Please indicate how the accuracy and quality control of the data was carried out.

Response: Thank you for this valid comment. Please we have explained how quality control of the data was carried out during the data collection stage and at the analysis stage. See page 4. 

2. Comment: Some of the independent variables in this study were selected by asking women for relevant information. Please indicate how to ensure the validity and reliability of the data.

Response: Thank you for your comment. We have explained how validity and reliability were maintained in the DHS data. Nonetheless, we have acknowledged that there is the possibility of social desirability and recall biasness as part of the limitations of the study. See page 17-18. 

3. Comment: Using STATA for a multilevel modelling may not be appropriate. Please explain the applicability of using STATA in this study.

Response: Thank you for your comment. Please the use of STATA for multi-level modelling has been explained in the analysis section. Again, STATA is the software but we used the ‘melogit’ which is a command in STATA for running the multilevel models. See page 6. 

Reviewer #2: 

4. Comment: After reading, I've got some questions/suggestions to the authors: There are too many lines in the tables, especially vertical lines, which do not conform to the statistical table specification.

Response: Thank you for your comment. Please we have formatted the Tables accordingly. See Table 1-3. 

5. Comment: The quality of DHS data should be mentioned in the paper where appropriate or explained by reference.

Response: We have mentioned the quality of the DHS data in the methods section of our paper. We have also specified this at the strength and limitation section. Thank you. See page 6 and 17-18. 

Thank you once more for your valuable comments.

---

## [Decision Letter · Decision Letter 1]

9 Dec 2020

Barriers to healthcare access and healthcare seeking for childhood illnesses among childbearing women in sub-Saharan Africa: A multilevel modelling of Demographic and Health Surveys

PONE-D-20-18533R1

Dear Dr. Budu,

We’re pleased to inform you that your manuscript has been judged scientifically suitable for publication and will be formally accepted for publication once it meets all outstanding technical requirements.

Kind regards,

Claudia Marotta

Academic Editor

PLOS ONE

Additional Editor Comments (optional):

Dear authors, congratulations

Reviewers' comments:

Reviewer's Responses to Questions

**Comments to the Author**

1. If the authors have adequately addressed your comments raised in a previous round of review and you feel that this manuscript is now acceptable for publication, you may indicate that here to bypass the “Comments to the Author” section, enter your conflict of interest statement in the “Confidential to Editor” section, and submit your "Accept" recommendation.

Reviewer #1: All comments have been addressed

Reviewer #2: All comments have been addressed

2. Is the manuscript technically sound, and do the data support the conclusions?

Reviewer #1: Yes

Reviewer #2: Yes

3. Has the statistical analysis been performed appropriately and rigorously? 

Reviewer #1: Yes

Reviewer #2: Yes

4. Have the authors made all data underlying the findings in their manuscript fully available?

Reviewer #1: Yes

Reviewer #2: Yes

5. Is the manuscript presented in an intelligible fashion and written in standard English?

Reviewer #1: Yes

Reviewer #2: Yes

6. Review Comments to the Author

Reviewer #1: (No Response)

Reviewer #2: The authors had adequately addressed my comments raised in last round of review and I felt that this manuscript is now acceptable for publication.

7. PLOS authors have the option to publish the peer review history of their article (what does this mean?). If published, this will include your full peer review and any attached files.

Reviewer #1: No

Reviewer #2: **Yes: **Zhang Yuhong

---

## [Editor Report · Acceptance letter]

28 Jan 2021

PONE-D-20-18533R1 

Barriers to healthcare access and healthcare seeking for childhood illnesses among childbearing women in sub-Saharan Africa: A multilevel modelling of Demographic and Health Surveys 

Dear Dr. Budu:

I'm pleased to inform you that your manuscript has been deemed suitable for publication in PLOS ONE. Congratulations! Your manuscript is now with our production department. 

Kind regards, 

on behalf of

Dr. Claudia Marotta 

Academic Editor

PLOS ONE